# Demonstration of a two-dimensional $\mathcal{PT}$-symmetric crystal

Mark Kremer [1], Tobias Biesenthal[1], Lukas J. Maczewsky[1], Matthias Heinrich[1], Ronny Thomale [2] & Alexander Szameit [1]

With the discovery of $\mathcal{PT}$-symmetric quantum mechanics, it was shown that even non-Hermitian systems may exhibit entirely real eigenvalue spectra. This finding did not only change the perception of quantum mechanics itself, it also significantly influenced the field of photonics. By appropriately designing one-dimensional distributions of gain and loss, it was possible to experimentally verify some of the hallmark features of $\mathcal{PT}$-symmetry using electromagnetic waves. Nevertheless, an experimental platform to study the impact of $\mathcal{PT}$-symmetry in two spatial dimensions has so far remained elusive. We break new grounds by devising a two-dimensional $\mathcal{PT}$-symmetric system based on photonic waveguide lattices with judiciously designed refractive index landscape and alternating loss. With this system at hand, we demonstrate a non-Hermitian two-dimensional topological phase transition that is closely linked to the emergence of topological mid-gap edge states.

---

[1] Institute for Physics, University of Rostock, Albert-Einstein-Straße 23, 18059 Rostock, Germany. [2] Department of Physics and Astronomy, Julius-Maximilians-Universität Würzburg, Am Hubland, 97074 Würzburg, Germany. Correspondence and requests for materials should be addressed to A.S. (email: alexander.szameit@uni-rostock.de)

In 1998, Carl M. Bender and Stefan Boettcher challenged the conventional wisdom of quantum mechanics that the Hamiltonian operator describing any quantum mechanical system has to be Hermitian[1]. They showed that Hamiltonians that are invariant under combined parity-time ($\mathcal{PT}$) symmetry transformations likewise can exhibit entirely real eigenvalue spectra[2]. This insight had a particularly profound impact in the field of photonics, where $\mathcal{PT}$-symmetric potential landscapes can be implemented by appropriately distributing gain and loss for electromagnetic waves[3–5]. Following this approach, it became possible to observe some of the hallmark features of $\mathcal{PT}$ symmetry, such as the existence of non-orthogonal eigenmodes[6] and exceptional points[7,8], diffusive coherent transport[9], and to study their implications in settings including nonlinearity[10], $\mathcal{PT}$-symmetric lasers[11,12], and topological phase transitions[13–15]. Similarly, $\mathcal{PT}$-symmetry has enriched other research fields ranging from $\mathcal{PT}$-symmetric atomic diffusion[16], superconducting wires[17,18], and $\mathcal{PT}$-symmetric electronic circuits[19].

Nevertheless, to this date, all experimental implementations of $\mathcal{PT}$-symmetric systems have been restricted to one effective spatial dimension, which is mostly due to technological limitations involved in realizing appropriate non-Hermitian potential landscapes.

In this work, we report on the experimental realization and characterization of a two-dimensional $\mathcal{PT}$-symmetric system by means of photonic waveguide lattices with judiciously designed refractive index landscape and alternating loss. A key result of our work is the demonstration of a non-Hermitian two-dimensional topological phase transition that coincides with the emergence of mid-gap edge states. Our findings lay the foundation for future investigations exploring the full potential of $\mathcal{PT}$-symmetric photonics in higher dimensions. Moreover, our approach may even hold the key for realizing two-dimensional $\mathcal{PT}$-symmetry beyond photonics, e.g., in matter waves and electronics.

## Results

**Theory.** $\mathcal{PT}$-symmetric systems are described by a Hamiltonian that is invariant under parity-time symmetry transformations[1]. In a more formal language, this means that if the Hamiltonian $\hat{H}$ commutes with the $\mathcal{PT}$-operator: $[\hat{H}, PT] = 0$, and the Hamiltonian shares the same set of eigenstates with the $\mathcal{PT}$-operator, then the entire set of eigenvalues of $\hat{H}$ is real. A necessary condition for this symmetry to hold is that the underlying potential obeys the relation $\hat{V}(-x) = \hat{V}*(x)$[1]. Whereas complex potentials tend to be difficult to realize in most physical systems, in 2007 it was shown that photonics provides a suitable testbed due to the complex-valued character of the refractive index[3,4]. Since then, $\mathcal{PT}$-symmetric systems have been explored in a variety of photonic platforms, ranging from waveguide arrays[6], fiber lattices[7], and coupled optical resonators[20] to plasmonics[21] and microwave cavities[22]. The implementation of $\mathcal{PT}$-symmetry in photonics is based on the observation that the Schrödinger equation of quantum mechanics for the probability amplitude $\psi$ $(x, y, t)$,

$$i\hbar \frac{\partial}{\partial t}\psi(x,y,t) = -\frac{\hbar^2}{2m}\nabla^2\psi(x,y,t) + V(x,y,t)\psi(x,y,t), \quad (1)$$

and the paraxial Helmholtz equation of electromagnetism for the electric field amplitude $E(x, y, z)$,

$$i\frac{n_0}{2k_0}\frac{\partial}{\partial z}E(x,y,z) = -\frac{n_0}{2k_0^2}\nabla^2 E(x,y,z) - n(x,y,z)E(x,y,z), \quad (2)$$

are formally equivalent if the potential $V(x, y, t)$ in the Schrödinger equation is replaced by the refractive index profile $-n(x, y,$ $z)$ in the Helmholtz equation[23]. $\mathcal{PT}$-symmetry then translates into the following condition for the complex refractive index:

$$n(-x,-y,z) = n^*(x,y,z) \quad (3)$$

In other words, the real part $\text{Re}(n(x, y, z))$ needs to follow a symmetric distribution, while the imaginary part $\text{Im}(n(x, y, z))$ has to be antisymmetric under the parity operation. In general, the latter implies that loss in one propagation direction has to be compensated by an identical gain in the opposite direction[3]. It turns out, however, that this stringent requirement can be relaxed in tight-binding systems, where an alternating loss distribution is sufficient to obtain $\mathcal{PT}$-symmetric behavior[9,24]. Indeed, such passive systems exhibit exactly the same evolution dynamics that one would expect in active structures if one accounts for a constant global loss by normalizing the field intensity, an approach we followed up on in our work.

Nevertheless, to date $\mathcal{PT}$-symmetry was only realized in one-dimensional (1D) systems, which drastically limits the spectrum of accessible physical effects. For example, the second spatial degree of freedom in two-dimensional (2D) structures allows for the study of various additional symmetries, such as rotation, and their interplay with $\mathcal{PT}$-symmetry. Moreover, one can introduce anisotropy with much more variety than in just one dimension. Furthermore, some physical effects fundamentally change their characteristics depending on the dimensionality, such as Anderson localization. This also applies to nonlinear optics, in particular solitons, which exhibit entirely different properties in 2D systems. The dimensionality is likewise of great importance for topological systems, since topological indices such as Chern number, Z2 invariant or Bott index in general necessitate at least a two-dimensional parameter space in which they can be defined. Moreover, chiral edge states can exist only along the edge of a 2D system. In our work, we break new grounds by devising a platform for the implementation of 2D $\mathcal{PT}$-symmetry that may enable future studies of this wide range of effects.

**Setting.** We consider so-called "photonic graphene", a regular arrangement of waveguides in a honeycomb geometry (sketched in Fig. 1a)[25]. In order to establish the necessary potential condition for $\mathcal{PT}$-symmetry Eq. (3), the two triangular sublattices of the honeycomb may exhibit different loss, symbolized by the yellow and blue filling of the individual lattice sites, respectively. We describe the light evolution in this system by the tight-binding approximation of Eq. (2), which reads as[26]:

$$i\partial_z a_{m,n} = i\gamma a_{m,n} + c\Big(b_{m-1,n} + b_{m,n+1} + b_{m,n-1}\Big) \quad (4a)$$

$$i\partial_z b_{m,n} = -i\gamma a_{m,n} + c\Big(a_{m+1,n} + a_{m,n+1} + a_{m,n-1}\Big) \quad (4b)$$

The $a_{m,n}$ and $b_{m,n}$ denote the amplitudes at the lattice sites of the two sublattices, $\gamma$ describes the gain/loss, and $c$ is the coupling constant.

Launching a light beam into the waveguides results in spatial beam dynamics (governed by Eqs. (4a) and (4b)) that, for the case of neither gain nor loss in both sublattices, resembles the evolution of a single electron in carbon-based graphene according to Eq. (1). One of the striking features of the graphene band structure is the existence of the so-called Dirac region in the vicinity of the conical intersection points ("diabolical points") between the first and the second bands, displayed in Fig. 1b. In these regions, the tight-binding Hamiltonian of our $\mathcal{PT}$-symmetric photonic graphene can be expanded into a Taylor series[26] to obtain a mathematical structure resembling the Dirac equation

$$\hat{H} = c\sqrt{3}\tilde{\nu}\sigma_1 + c\sqrt{3}\tilde{\mu}\sigma_2 + i\gamma\sigma_3 \quad (5)$$

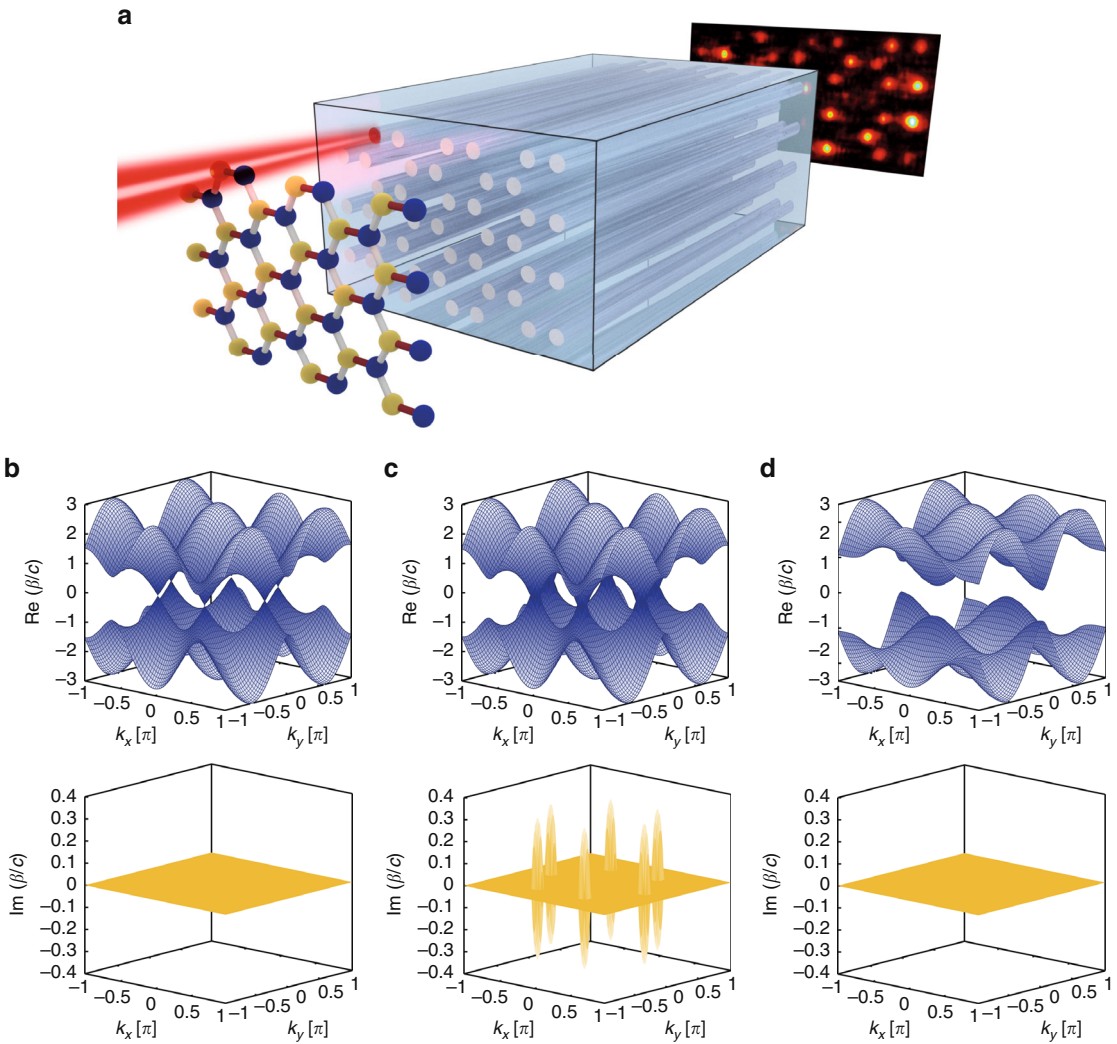

**Fig. 1** The $\mathcal{PT}$-symmetric graphene lattice. **a** Due to the quantum-optical analogy between solids and waveguide lattices, the probability amplitude of a single electron in a carbon graphene lattice shares the same evolution equation as a light beam injected into a honeycomb waveguide lattice. **b** The dispersion relation of a graphene lattice with no gain/loss and no strain ($\tau = 1$), **c** with gain/loss and no strain ($\tau = 1$), and **d** with gain/loss above the critical point ($\tau > 2 + \frac{\gamma}{c}$), which corresponds to the unbroken $\mathcal{PT}$-symmetry regime

that describes relativistic quantum particles. Here, $\sigma_{1,2,3}$ are the Pauli matrices:

$$\sigma_1 = \begin{pmatrix} 0 & 1 \\ 1 & 0 \end{pmatrix}; \sigma_2 = \begin{pmatrix} 0 & -i \\ i & 0 \end{pmatrix}; \sigma_3 = \begin{pmatrix} 1 & 0 \\ 0 & -1 \end{pmatrix}; \quad (6)$$

$2\gamma$ denotes the difference in loss between the sublattices, and $c$ is the intersite hopping. The quantities $\tilde{\mu}$ and $\tilde{\nu}$ represent the components of the transverse wave vector $k_x$, $k_y$ measured from the position of the original Dirac point. For simplicity, we suppressed an additional term $-i\Gamma\sigma_0$ that arises from the passive nature of our system, where $\Gamma$ is the average loss of both sublattices, and $\sigma_0$ is the unity matrix. The non-Hermitian Hamiltonian in Eq. (5) exhibits a complex dispersion relation with a non-real eigenvalue spectrum[27]. Mathematically, complex eigenvalues of the Hamiltonian appear whenever the $\mathcal{PT}$-operator and the Hamiltonian cease to share all of their eigenvectors. Such a system is said to have broken $\mathcal{PT}$-symmetry, although the $\mathcal{PT}$-operator still commutes with the Hamiltonian. This seeming paradox stems from the fact that the $\mathcal{T}$-operator is anti-linear. A graph of the real part of the graphene dispersion relation

with $\frac{\gamma}{c} = 0.32$ is shown in Fig. 1c. In contrast to "conventional" (i.e., dissipation-less) graphene, the real part of the dispersion relation is now a single-sheeted hyperboloid. The lower part of Fig. 1c shows the imaginary part of the dispersion relation, highlighting the purely imaginary eigenvalues around the original vertices.

One can drive the system back into the unbroken $\mathcal{PT}$-symmetry regime by applying a linear strain $\tau$, where $\tau = 1$ corresponds to the unstrained case. This strain is applied as indicated in Fig. 1a by the red connections between the atoms. In Hermitian lattices ($\gamma = 0$), increasing the strain pushes pairs of Dirac points towards one another until they merge at $\tau = 2$[28]. For any given loss factor $\gamma$, all eigenvalues of the system become real above a threshold strain $\tau \geq 2 + \frac{\gamma}{c}$[26]. Therefore, in such a setting, the structure can exhibit unbroken $\mathcal{PT}$-symmetry, with the transition occurring exactly at $\tau = 2 + \frac{\gamma}{c}$. The Hamiltonian of this system reads as[26]

$$\hat{H} = \left[ c(\tau - 2) - \frac{3}{2} ct\tilde{\mu}^2 + c\tilde{\nu}^2 \right] \sigma_1 + ct\sqrt{3}\tilde{\mu}\sigma_2 + i\gamma\sigma_3, \quad (7)$$

**a**

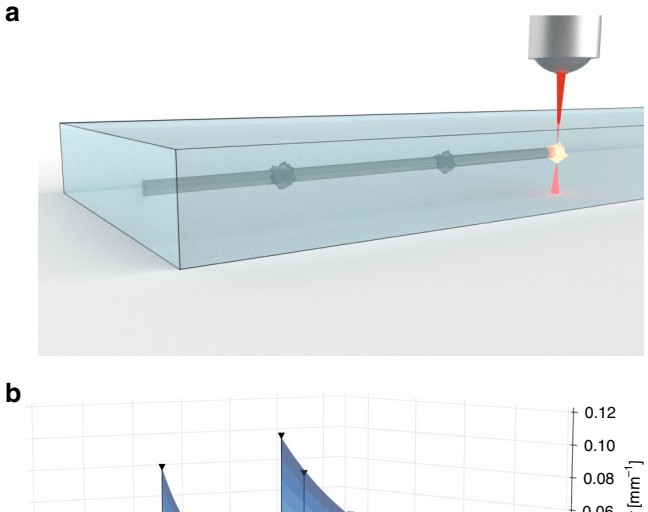

**b**

**Fig. 2** Realizing artificial loss. **a** Schematic of the artificial scatterers introduced during the fabrication process by dwelling. **b** Experimentally obtainable artificial losses as a function of dwelling time and separation of the scatter centers

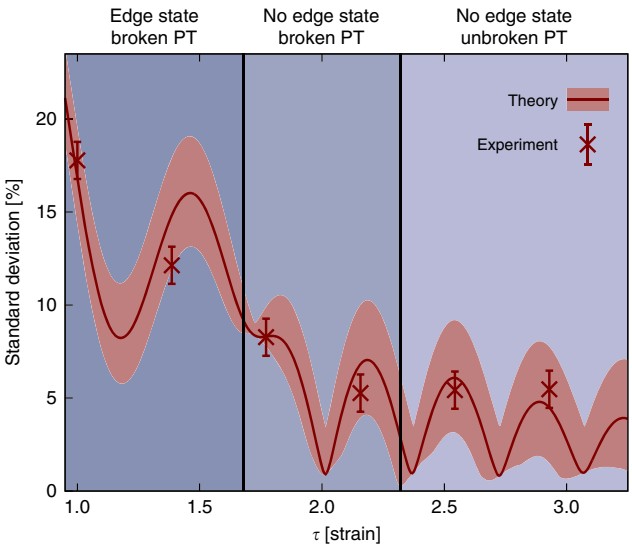

**Fig. 3** Demonstration of the phase transition from broken to unbroken $\mathcal{PT}$-symmetry. The standard deviation of the output intensity pattern resulting from 6 excitations (3 unit cells). In the unbroken $\mathcal{PT}$-symmetric phase, this quantity approaches zero. The simulations are performed by using the Schrödinger equation in the tight binding limit Eqs. (4a) and (4b). The red shaded area represents the calculated uncertainty arising from possible unequal incoupling in the experiment, whereas the error bars account for fluctuations of the measured data. The blue shaded regions show the three distinct phases associated with the strain and loss parameters of our structure (see Fig. 4a)

with $\Delta = c(\tau - 2)$ denoting the gap in the spectrum[26]. In Fig. 1d, we show the real and imaginary parts of the dispersion relation for photonic graphene with the same loss factor as in Fig. 1c in the presence of a strain given by $\tau = 2 + \frac{\gamma}{c} + 0.61$. Evidently, all eigenvalues are real, and a gap has opened in accordance with Eq. (7).

**Experimental methods**. In order to implement the $\mathcal{PT}$-symmetric photonic graphene lattice, we employ the direct laser-writing technology[29]. The desired loss in the system is realized by introducing a certain concentration of microscopic scattering points along the waveguides by dwelling, as sketched in Fig. 2a. As both the dwelling time and the separation between the individual scattering points can be freely tuned, this approach allows for a wide range of artificial losses to be chosen without compromising the real part of the refractive index or introducing directionality into the system. Figure 2b shows our calibration measurements of the realized loss, as a function of dwelling time and scattering point separation. The general trends are clearly visible: the cumulative loss experienced by light propagating through the waveguides systematically increases the smaller the separations and the longer the dwelling times. The strain is realized by reducing the distance between the waveguides with the red connection bond, shown in Fig. 1a, while the distances between the waveguides with the gray connection bonds are kept constant.

**$\mathcal{PT}$-phase transition**. With these tools at hand, we can now proceed to experimentally demonstrate the $\mathcal{PT}$-symmetry transition in our 2D photonic graphene lattice. When no strain is present, $\mathcal{PT}$-symmetry is broken and, hence, the spectrum is complex. After a certain propagation distance, the remaining light tends to reside predominantly in the lossless sublattice[30]. As the strain is increased, however, the system enters the regime

of unbroken $\mathcal{PT}$-symmetry, resulting in a real spectrum. This transition can be readily visualized by exploiting the fact that in the broken $\mathcal{PT}$-phase, the eigenvalues exhibit a wide range of different imaginary parts, and the power remaining in the lattice depends strongly on the injection site. In contrast, the unbroken $\mathcal{PT}$-phase is by definition characterized by all eigenvalues having the same imaginary part and, for an infinite system, the power decay in the lattice becomes entirely independent of the excited waveguide. Therefore, as the strain is increased above the critical value of $\tau = 2 + \frac{\gamma}{c}$, the standard deviation of the transmitted power will eventually vanish[14]. Furthermore, one has to take into account that the modes are non-orthogonal even in the unbroken $\mathcal{PT}$-regime, hence the power is not preserved[3] and one does not observe a sharp drop of the standard deviation at the transition point. We demonstrate this behavior by fabricating 6 samples with $\gamma = 0.15\,\text{cm}^{-1}$, a coupling of $c = 0.475\,\text{cm}^{-1}$ and strains ranging from $1 \leq \tau \leq 2.9$. In each sample, we perform 6 single-channel excitations of bulk sites, corresponding to 3 unit cells, and measure the total power remaining in the lattice at the output facet. The extracted data is plotted in Fig. 3. As expected, the variance substantially decreases and tends toward zero as the lattice is brought into its unbroken phase.

**Topological phase transition**. The phase transition from the broken to the unbroken $\mathcal{PT}$-symmetry regime in the graphene lattice is inextricably linked to a topological phase transition related to the emergence of topological mid-gap states. Figure 4a summarizes this feature in a $\tau$–$\gamma$ phase diagram. The presence or absence of topological mid-gap states at the edge of the Hermitian graphene lattice ($\gamma = 0$) can be reconciled from the perspective of the winding number in a Su–Shrieffer–Heeger (SSH) chain perpendicular to the edges[31,32]. A topological phase transition from a

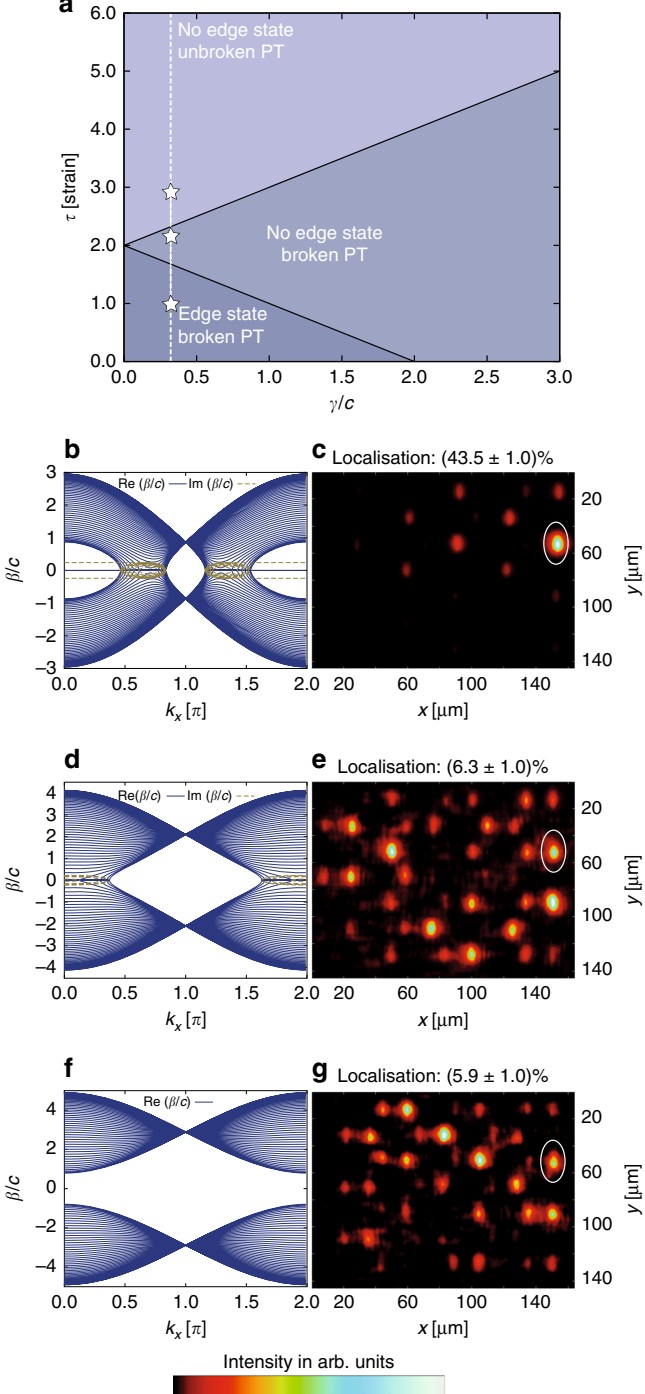

**Fig. 4** Demonstration of the topological transition in the $\mathcal{PT}$-symmetric structure. **a** The $\tau$–$\gamma$ phase diagram shows three distinct phases associated with the strain and loss parameters of our structure. The three stars correspond to the parameters used in (**b**)–(**g**). **b** For a strain of $\tau < 2 - \frac{\gamma}{c}$, the system is in the broken $\mathcal{PT}$-symmetric phase and topological mid-gap states are present. In the left panel, the corresponding dispersion relation is shown. **c** Experimental image at the sample output facet, where light was injected into the marked edge waveguide and remains close to the edge as the mid-gap state exists. The ratio of light in the marked waveguide in proportion to the intensity of all waveguides is about 43.5%. **d** For an intermediate strain $2 - \frac{\gamma}{c} < \tau < 2 + \frac{\gamma}{c}$, the system is still in the broken $\mathcal{PT}$-symmetric phase, but the topological mid-gap states cease to exist. In the left panel, the corresponding dispersion relation is shown. **e** Experimental image at the sample output facet, where light was injected into the marked edge waveguide and spreads into the bulk as no mid-gap state exists. The ratio of light in the marked waveguide in proportion to the intensity of all waveguides is about 6.3%. **f** For sufficiently strong strain $\tau > 2 + \frac{\gamma}{c}$, the system is finally driven into the unbroken $\mathcal{PT}$-symmetric phase, where no topological mid-gap states exist either. In the left panel, the corresponding dispersion relation is shown. **g** Experimental image at the sample output facet, where light was injected into the marked edge waveguide and spreads into the bulk as no mid-gap state exists. The ratio of light in the marked waveguide in proportion to the intensity of all waveguides is about 5.9%

$\mathcal{PT}$-symmetry is invariably broken—yet, as long as the hyperboloids in the real part of the dispersion relation still touch, topological mid-gap states are prevented from forming at the bearded edge[36]. This can be intuitively explained by the shape of these states, which are known to reside exclusively within one of the sublattices, and, hence, experience solely $+\gamma$ or $-\gamma$. As a consequence, the imaginary part of the mid-gap dispersion is $\pm\gamma$ and, from the perspective of a $\mathcal{PT}$-symmetric SSH chain, this implies the disappearance of the mid-gap state for $\tau \geq 2 - \gamma/c$[14,36]. Therefore, when starting in the topologically non-trivial domain with mid-gap states and broken $\mathcal{PT}$-symmetry, and following a vertical trajectory (fixed $\gamma$ and increasing $\tau$) in the phase diagram, the system passes not one but two phase transitions. The first one is of topological nature and occurs when the direct gap of the topological mid-gap states closes at $\tau = 2 - \gamma/c$. The second occurs at $\tau = 2 + \gamma/c$ when the gapped unbroken $\mathcal{PT}$-symmetric domain is reached (see Fig. 4a).

These transitions are exactly what we observe in our experiment. For the unstrained system ($\tau = 1$) and a loss of $\gamma = 0.15\,\mathrm{cm}^{-1}$ such that $\frac{\gamma}{c} = 0.32$, the systems exhibits topological mid-gap states (Fig. 4b), which we excite by launching light into an edge waveguide (Fig. 4c). Upon increasing the strain to $\tau = 2.2$, the mid-gap edge states disappear, in the dispersion relation (Fig. 4d) as well as in experiment (Fig. 4e). However, the system is still in the broken $\mathcal{PT}$-symmetric phase, as shown in Fig. 3b. By further increasing the strain to $\tau = 2.9$, we finally drive the system into the unbroken $\mathcal{PT}$-symmetric phase, whereas the mid-gap states are likewise absent (see Fig. 4f for the dispersion relation and Fig. 4g for the experimental data). Our results clearly show the close links between $\mathcal{PT}$-symmetric and topological phase transitions, and their related transition points as shown in Fig. 4a. This rich interplay stems from the fact that it is the very existence of a band gap that allows for the proper definition of a topological invariant in this structure.

2D topological semimetal to a trivial insulator, accompanied by a change of the SSH winding number, takes place at $\tau = 2$[28,33,34]. For any $\gamma > 0$, however, a topological mid-gap state spontaneously breaks $\mathcal{PT}$-symmetry, since its real dispersive part is pinned. As a consequence, the unbroken $\mathcal{PT}$-symmetric domain of the $\tau$–$\gamma$ phase diagram does not exhibit any edge modes, as shown in Fig. 4a. In other words, one can either observe unbroken $\mathcal{PT}$-symmetry, or topological mid-gap states, but never both at the same time, since the two phenomena are mutually exclusive[35]. Interestingly, a third domain is wedged between the previously discussed cases in the phase diagram. It arises when the strain $\tau$ exceeds the gap threshold determined by $\gamma$. As the gap is closed,

## Discussion
In our work, we devised and experimentally demonstrated a 2D $\mathcal{PT}$-symmetric crystal, using an optical platform. To this end, we

developed a technology to efficiently introduce artificial losses into the system and unequivocally proved that our realized structure is indeed in the unbroken $\mathcal{PT}$-symmetric phase. Moreover, we highlighted the close connection of a $\mathcal{PT}$-symmetry phase transition to a topological phase transition in our graphene lattice. These findings lay the foundations for realizing 2D $\mathcal{PT}$-symmetry in other wave systems beyond photonics, such as matter waves, sound waves, and possibly even plasmonics and electronic circuits. Moreover, our work opens the gate for future investigations exploring the full potential of $\mathcal{PT}$-symmetry in higher dimensions and may provide the tools to experimentally address numerous exciting questions such as the impact of nonlinearity, single photon interference, and many-body effects in 2D $\mathcal{PT}$-symmetric systems.

## Methods

**Fabrication of the structures**. The waveguides were manufactured using the femtosecond laser writing method[29] in 10 cm long samples composed of fused silica glass (Corning 7980). The laser pulses are created by a regenerative Ti: Sapphire amplifier system (Coherent RegA 9000 seeded with a Mira 900) and exhibit an energy of 450 nJ @ 800 nm wavelength and 100 kHz repetition rate. A nano-positioning system in conjunction with a 20× microscope objective (0.35 NA) provides the highly accurate focusing of the laser beam 50–800 μm under the sample surface. By translating the sample with a speed of about 100 mm/min, the refractive index at the focal point is increased by approximately $7 \times 10^{-4}$, resulting in waveguides with a mode field diameter of 10.4 μm × 8 μm for the 633 nm illumination wavelength. Intrinsic propagation losses and birefringence were estimated to be 0.2 dB cm$^{-1}$ and $10^{-6}$, respectively.

**Characterization of the structures**. The single lattice site excitations were performed with light of 633 nm from a Helium–Neon laser (Melles-Griot, 35 mW). In order to focus into the sample a 10× microscope objective (0.25 NA) was used. The output facet was imaged with another 10× microscope objective onto a CCD camera (Basler Aviator). Upon characterization, the output intensity patterns resulting from the excitation of lossy sites were normalized to compensate for the systematically lower injection efficiency.

## Data availability

All experimental data and any related experimental background information not mentioned in the text are available from the authors on reasonable request.

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

## Acknowledgements

A.S. gratefully acknowledges the financial support from the Deutsche Forschungsgemeinschaft (Grants SZ 276/9-1, SZ 276/19-1, and SZ 276/20-1) and the Alfried Krupp von Bohlen und Halbach Foundation. R.T. is supported by DFG-SFB 1170 (Project B04) and ERC-StG-Thomale-TOPOLECTRICS-336012. The authors would also like to thank C. Otto for preparing the high-quality fused silica samples used in all the experiments presented here. A.S. and R.T. are supported by DFG-EXC 2471 „ct.qmat".

## Author contributions

M.K. and A.S. developed the idea and designed the structure. M.K., R.T., and A.S. worked out the theory. M.K., T.B., and L.J.M. fabricated the samples and performed the measurements. A.S. supervised the project. All authors discussed the results and co-wrote the paper.

## Additional information

**Competing interests:** The authors declare no competing interests.

