## [Peer Review File · Nature Communications]

Reviewers' Comments:

Reviewer #1:

Remarks to the Author:

The authors experimentally create a two-dimensional PT symmetric lattice and use it to demonstrate the existence of a non-Hermitian two-dimensional topological phase transition that coincides with the emergence of mid-gap edge states. PT symmetric physics is an important and active research area; in this regard, the present manuscript contributes positively.

I find the results reported in the manuscript rather "thin". The authors did not explain why their key result (the demonstration of a non-Hermitian two-dimensional topological phase transition that coincides with the emergence of mid-gap edge states) is very important that warrant publication in Nature Communications? The authors did not explain the physical ramification of such a phase transition. Unless the authors make a clear case explaining why the experimental results reported in this manuscript are very important, I cannot recommend publication of this manuscript in Nature Communication in its current form.

Below, I outline few technical issues:

1. The authors start their abstract by stating "In 1998, Carl Bender challenged the perceived wisdom" and then cite Ref[1]. Then they go on and write "He showed that, Hamiltonians that are invariant", again, citing Ref.[2,3]. First of all, Ref. [1] is written by two authors and NOT by Carl Bender alone! Therefore, the authors need to change "In 1998, Carl Bender challenged ..." to "In 1998, Carl Bender and S. Boettcher challenged ...". Second, Ref.[2] is written by three authors and NOT Carl Bender alone, so why the authors write "He showed ..."? This type of scientific practice (discrediting some authors) is not acceptable!
2. line 51 (above Eq.(1)), the authors need to replace "density" with "amplitude".
3. Below Eq.(2) the authors write the potential as $V(x)$ whereas in Eq.(1) the potential is a function of x, y and t , i.e., $V(x,y,t)$. The authors need to be consistent in their notation.
4. In Eq.(3) the authors write the PT condition for the refractive index n as $n(x)=n^*(-x)$. But, In Eq.(2) the refractive index is a function of x,y and z , i.e., $n(x,y,z)$. Again, the authors need to be consistent.
5. The theoretical part seems to be done on Hamiltonians that are given by Eq.(4) or (6). I would like to see simulations performed on Eq.(2) using honeycomb refractive index.
6. When talking about PT photonics, the authors need to cite the two papers: Z. Musslimani, K. Makris, R. El Ganainy, and D. Christodoulides, Phys. Rev. Lett. 100, 030402 (2008) and R. El Ganainy, K. Makris, M. Khajavikhan, Z. Musslimani, S. Rotter and D. Christodoulides, "Non-Hermitian physics and PT symmetry", Nature Physics, 14, 11, (2018)

Reviewer #2:

Remarks to the Author:

The submitted paper presents the experimental realization and characterization of a two-dimensional PT-symmetric lattice (photonic graphene) and discusses in details its phase transitions, topological properties, and emergence of edge states. While the theory of non-Hermitian topological systems is not new – it is currently the subject of extensive research efforts by many groups – the paper provides an elegant experimental demonstration of several important

concepts and phenomena that have been predicted to emerge in this type of photonic platforms. For example, the authors developed a new technology to efficiently and controllably introduce artificial losses into the system, and exploited it to demonstrate non-Hermitian two-dimensional topological phase transitions. Overall, the paper is convincing, well written, and accessible to a broad audience. There are, however, several important issues that should be addressed, as discussed in the following.

In summary, for the reasons outlined above, I support the publication of the submitted manuscript provided that the following points are properly addressed:

(1) In the abstract, the authors mention "non-reciprocal light evolution [7]" as one of the "hallmark features of PT symmetry". However, the system studied in [7] is perfectly reciprocal! Nonreciprocity is generally unrelated to PT symmetry, as the presence of loss and gain does not break Lorentz reciprocity theorem for electromagnetic waves. Indeed, the scattering matrix of any linear PT-symmetric system is symmetric ($S_{12}=S_{21}$). The authors should remove or fix this statement.

(2) The labels of Fig. 1b,c,d, Fig. 2b, and Fig. 4b,c,d are too small. Also, the authors should explicitly define the symbols in these figures.

(3) Under "Additional information", the authors write "Supplementary information is available in the online version of the paper." However, no supplementary document was provided.

(4) It is not clear how the "strain" is implemented experimentally. The author should clarify this point, perhaps also including images of their fabricated samples.

(5) Two additional recent papers that are relevant to the topic of non-Hermitian topological systems are: S. A. Hassani Gangaraj and F. Monticone, "Topological Waveguiding near an Exceptional Point: Defect-Immune, Slow-Light, and Loss-Immune Propagation," Phys. Rev. Lett. 121, 093901 (2018); M. Li, X. Ni, M. Weiner, A. Alù, and A. B. Khanikaev, "Topological phases and nonreciprocal edge states in non-Hermitian Floquet Insulators," arXiv:1807.00913v1 (2018).

(6) The authors write: "In contrast, in the unbroken PT-phase all eigenvalues exhibit the same imaginary part and, for an infinite system, the power decay in the lattice were independent of the excited waveguide. Therefore, as the strain is increased above the critical value, the standard deviation of the transmitted power will eventually vanish [12]." Since this is key to the demonstration of a phase transition, the authors should elaborate more on this point. Also, they should explain how the theoretical results in Fig. 3 have been calculated.

(7) I was expecting to see a more drastic difference between the intensity plots in Fig. 4c and Fig. 4d, right panels, since a phase transition occurs between them. Can the authors clarify why these intensity plots look so similar? Also, these panels need a colorbar.

(8) A typo: "the full potential of PT-symmetric in higher dimensions" -> "PT symmetry" or "PT-symmetric systems"

Reviewer #3:

Remarks to the Author:

In this manuscript entitled "Demonstration of a two-dimensional PT-symmetric crystal: Bulk dynamics, topology, and edge states", the authors reported their experimental demonstration of PT-symmetry phase transition and topological phase transition in a photonic waveguide lattice. The waveguide lattice is fabricated in fused silica glass using laser direct writing technique, where parameters of the system, e.g. additional loss, coupling strength, strain strength, lattice structure,

can be adjusted. By judiciously choosing parameters, a two-dimensional PT-symmetric waveguides lattice with honeycomb structure is achieved. This PT-symmetric system not only has broken/unbroken PT-symmetry phase transition, but also has topological phase transition due to its special band structure. The merit of this work lies in the 2D lattices which could demonstrate some features beyond what's observed in 1D PT-symmetric structures. Overall the manuscript is well organized. It could be improved if the following issues are addressed or corrected:

1. In Eq. (2), it seems that the unit of the term on the left hand side does not match with that of the right hand side.
2. Eq. (1) and Eq. (2) set up the framework to study PT symmetry in two-dimensional photonic crystals, based on which the constraints on the complex refractive index can be derived. Yet in the following discussion, the condition of PT symmetry is given with respect to a one-dimensional optical potential, as shown in Eq. (3). It will be helpful if the authors could provide more discussion on the condition of PT symmetry for a two-dimensional system.
3. There is a lack of argument or demonstration explaining why the refractive index potential in this two-dimensional photonic graphene lattice satisfies the condition of PT symmetry.
4. In Fig. 3, the number of data points provided is not enough to show a complete trend in which the standard deviation of the output intensity changes. In particular, more data points are needed to show the eventually vanishing behaviour when the system is in the unbroken PT regime
5. In Fig. 3, the experimental data seem to have a slight deviation from the theory. In theory, the standard deviation does not exhibit a sharp decrease to zero after the phase transition point ($\tau=2.316$ as derived from the parameters provided). But in the experiment, the change appears to be abrupt around the phase transition point. The reason for this difference should be provided.
6. In this experiment, the PT-symmetric optical potential is realized with an alternating loss, and thus the system exhibits an overall lossy feature. How is the topological phase transition point, as mentioned on page 6, influenced by the global loss of the system?
7. On page 7, the authors state that "Moreover, we highlighted the close connection of a PT-symmetry phase transition to a topological phase transition in our graphene lattice." Except for the conventional argument that "the topological mid-gap state spontaneously breaks PT-symmetry" (page 6), it is not apparent how these two kinds of phase transitions are related. The link between them should be clarified.
8. It will be helpful if the authors could provide more information on the advantageous features and applications of the two-dimensional PT-symmetric crystal compared to the previous one-dimensional platforms with PT symmetry.
9. When this system undergoes a PT-symmetry phase transition, it is theoretically predicted that a topological phase transition will be accompanied. More interesting, when the loss difference $\gamma > 0$, these two transitions can be observed at different values of strain τ , which is shown in Fig. 4. But the observed results in the right panel of Fig. 4(b)-(d) are not very clear. The paper says in Fig. 4(c) & 4(d) the light injected is spreading into the bulk due to the absence of edge states, and in Fig. 4(b) the edge states exist thus light stays close to the edge. However, the lattice structure presented in this manuscript only has three sublattices, thus the definitions of "edge part" and "bulk region" of the system is not very clear. In Fig. 4(b), it does not seem like the light is just "remaining at the edge", one can also see some light penetrates into the bulk. In order to convincingly state that the topological edge states are present, is there a better way to visualize and quantify this effect? It is important to clarify this issue in Fig. 4, because "the demonstration of a non-Hermitian two-dimensional topological phase transition" is a key result of this work, as said in the abstract. It seems a larger system with more sublattices will make it clear that the light

in the waveguides is really confined to the edge.

10 There are some typos in the manuscript. For example, on the left hand side of Eq. (1), $\partial/\partial z$ should be corrected as $\partial/\partial t$, i.e., the partial z should be partial t. In Figure caption 4, "(b)" is repeated twice where "(c)" and "(d)" should be used.

Reviewers' comments:

Reviewer #1 (Remarks to the Author):

Comment:

The authors experimentally create a two-dimensional PT symmetric lattice and use it to demonstrate the existence of a non-Hermitian two-dimensional topological phase transition that coincides with the emergence of mid-gap edge states. PT symmetric physics is an important and active research area; in this regard, the present manuscript contributes positively.

Response:

We thank the reviewer for this evaluation and are happy to see that he/she recognises our manuscript as part of an active and important field of research.

Comment:

I find the results reported in the manuscript rather "thin". The authors did not explain why their key result (the demonstration of a non-Hermitian two-dimensional topological phase transition that coincides with the emergence of mid-gap edge states) is very important that warrant publication in Nature Communications? The authors did not explain the physical ramification of such a phase transition. Unless the authors make a clear case explaining why the experimental results reported in this manuscript are very important, I cannot recommend publication of this manuscript in Nature Communication in its current form.

Response:

The reviewer himself/herself pointed out earlier that *PT symmetric physics is an important and active research area*. In this vein, we aim to contribute to this exciting field by introducing and characterising an experimental platform that allows verification of numerous recent theoretical proposals. To this date, there is no experimental platform available to study and verify PT-symmetric physics in two spatial dimensions. In order to show that we offer a suitable and versatile platform we study bulk and edge effects and the interplay of PT-symmetry and topology. With this palette of opportunities at hand we pave the way for a plethora of future research. Let us mention a few contexts in which the study of 2D PT structures will provide access to exciting physical effects:

The first is the extended degrees of freedom in designing 2D PT-symmetric lattice structures. In comparison to 1D systems, one can explore many more symmetries, like rotational symmetries, and their interplay with PT-symmetry. On the other hand, one can introduce anisotropy with much more variety than in just one dimension. Beyond this, one can experience completely different behavior of certain physical effects, based on their dimensionality. A famous example is Anderson localization, where the localization depends on the number of spatial dimensions. It is furthermore fundamental for (artificial) magnetic fields to have at least two spatial dimensions. This is of particular interest since (artificial) magnetic fields have proven to be an essential ingredient e.g. when dealing with, Landau levels, Bloch oscillations, or topological insulators.

The second is about nonlinear effects. In particular, solitons behave fundamentally different in two-dimensional systems, e.g. they may exhibit a threshold unlike in 1D. With the introduced toolkit, one can experimentally explore what happens to moving solitons, collisions, or modulation instabilities, when Hermitian physics is extended to the realm of PT-symmetric Hamiltonians.

The third, perhaps most important reason is that topological insulators can be implemented only in two or dimensions: in 1D, neither a Chern number, a Z2 invariant nor a Bott index can be defined. Moreover, chiral edge states can exist only along the edge of a 2D system. Having a PT-symmetric topological insulator – a key goal in physics today – is possible only in a 2D PT-symmetric system.

Comment:

1. The authors start their abstract by stating "In 1998, Carl Bender challenged the perceived wisdom" and then cite Ref[1]. Then they go on and write "He showed that, Hamiltonians that are invariant", again, citing Ref.[2,3]. First of all, Ref. [1] is written by two authors and NOT by Carl Bender alone! Therefore, the authors need to change "In 1998, Carl Bender challenged ..." to "In 1998, Carl Bender and S. Boettcher challenged ...". Second, Ref.[2] is written by three authors and NOT Carl Bender alone, so why the authors write "He showed ..."? This type of scientific practice (discrediting some authors) is not acceptable!

Response:

We wholeheartedly agree with this comment and apologize for this oversight.

Comment:

2. line 51 (above Eq.(1)), the authors need to replace "density" with "amplitude".

Response:

We agree and corrected this in the manuscript.

Comment:

3. Below Eq.(2) the authors write the potential as $V(x)$ whereas in Eq.(1) the potential is a function of x, y and t , i.e., $V(x,y,t)$. The authors need to be consistent in their notation.

Response:

We agree and changed this in the manuscript.

Comment:

4. In Eq.(3) the authors write the PT condition for the refractive index n as $n(x)=n^*(-x)$. But, In Eq.(2) the refractive index is a function of x,y and z , i.e., $n(x,y,z)$. Again, the authors need to be consistent.

Response:

We agree and changed this in the manuscript.

Comment:

5. The theoretical part seems to be done on Hamiltonians that are given by Eq.(4) or (6). I would like to see simulations performed on Eq.(2) using honeycomb refractive index.

Response:

The simulations are indeed performed by using the Schrödinger equation, in the tight-binding limit (as in A. Szameit, M. C. Rechtsman, O. Bahat-Treidel, and M. Segev, "PT-Symmetry in honeycomb photonic lattices," Phys. Rev. A 84(2), 021806(R) (2011).) Therefore, the whole symmetry of the lattice is taken into account. Furthermore, many works have shown that it is sufficient to use the tight binding equation for the description of waveguide arrays, instead of a full numerical model. We indeed did not mention the tight-binding Schrödinger equation explicitly in the previous version of the manuscript and gave only a reference to the above-mentioned paper. This is corrected in the revised version: The tight-binding Schrödinger equation as the basis for our simulations is now explicitly given as Eq. (4).

Comment:

6. When talking about PT photonics, the authors need to cite the two papers: Z. Musslimani, K. Makris, R. El Ganainy, and D. Christodoulides, Phys. Rev. Lett. 100, 030402 (2008) and R. El Ganainy, K. Makris, M. Khajavikhan, Z. Musslimani, S. Rotter and D. Christodoulides, "Non-Hermitian physics and PT symmetry", Nature Physics, 14, 11, (2018)

Response:

We agree and have included references to these papers.

Reviewer #2 (Remarks to the Author):**Comment:**

The submitted paper presents the experimental realization and characterization of a two-dimensional PT-symmetric lattice (photonic graphene) and discusses in details its phase transitions, topological properties, and emergence of edge states. While the theory of non-Hermitian topological systems is not new – it is currently the subject of extensive research efforts by many groups – the paper provides an elegant experimental demonstration of several important concepts and phenomena that have

been predicted to emerge in this type of photonic platforms. For example, the authors developed a new technology to efficiently and controllably introduce artificial losses into the system, and exploited it to demonstrate non-Hermitian two-dimensional topological phase transitions. Overall, the paper is convincing, well written, and accessible to a broad audience. There are, however, several important issues that should be addressed, as discussed in the following.

Response:

We thank the reviewer for his time and effort to read and evaluate the manuscript. Furthermore we appreciate that the reviewer that he shares our opinion about the relevance and potential impact of the manuscript.

Comment:

In summary, for the reasons outlined above, I support the publication of the submitted manuscript provided that the following points are properly addressed:

Response:

We thank the referee for this assessment.

Comment:

(1) In the abstract, the authors mention “non-reciprocal light evolution [7]” as one of the “hallmark features of PT symmetry”. However, the system studied in [7] is perfectly reciprocal! Nonreciprocity is generally unrelated to PT symmetry, as the presence of loss and gain does not break Lorentz reciprocity theorem for electromagnetic waves. Indeed, the scattering matrix of any linear PT-symmetric system is symmetric ($S_{12}=S_{21}$). The authors should remove or fix this statement.

Response:

We agree that this statement is potentially misleading and, hence, have removed it.

Comment:

(2) The labels of Fig. 1b,c,d, Fig. 2b, and Fig. 4b,c,d are too small. Also, the authors should explicitly define the symbols in these figures.

Response:

We agree and have redesigned all necessary figure parts.

Comment:

(3) Under “Additional information”, the authors write “Supplementary information is available in the online version of the paper.” However, no supplementary document was provided.

Response:

We apologize for this flaw and removed the statement.

Comment:

(4) *It is not clear how the “strain” is implemented experimentally. The author should clarify this point, perhaps also including images of their fabricated samples.*

Response:

We regret that this detail was not presented more clearly and have added a statement in the manuscript for clarification. The strain is implemented by decreasing the distance between the waveguides with the red connection bond, shown in Figure 1a, while the distances between the waveguides with the grey connection bonds are kept constant.

Comment:

(5) *Two additional recent papers that are relevant to the topic of non-Hermitian topological systems are: S. A. Hassani Gangaraj and F. Monticone, "Topological Waveguiding near an Exceptional Point: Defect-Immune, Slow-Light, and Loss-Immune Propagation," Phys. Rev. Lett. 121, 093901 (2018); M. Li, X. Ni, M. Weiner, A. Alù, and A. B. Khanikaev, "Topological phases and nonreciprocal edge states in non-Hermitian Floquet Insulators," arXiv:1807.00913v1 (2018).*

Response:

We agree and now cite those papers.

Comment:

(6) *The authors write: “In contrast, in the unbroken PT-phase all eigenvalues exhibit the same imaginary part and, for an infinite system, the power decay in the lattice were independent of the excited waveguide. Therefore, as the strain is increased above the critical value, the standard deviation of the transmitted power will eventually vanish [12].” Since this is key to the demonstration of a phase transition, the authors should elaborate more on this point. Also, they should explain how the theoretical results in Fig. 3 have been calculated.*

Response:

When light is launched into different waveguides, different ensembles of eigenmodes are excited. If one assumes a system with gain and loss in the broken PT regime, each of these eigenmodes experiences either gain or loss according to the imaginary parts of their corresponding eigenvalues. When all eigenvalues become real in the unbroken PT regime, all modes experience neither loss nor gain, and the standard deviation eventually drops to zero. Furthermore, one has to take into account that the modes are not orthogonal even in the unbroken PT-regime, hence the power is not preserved [3] and one does not observe a sharp drop of the standard deviation at the transition point. We added the last paragraph in the manuscript in order to clarify this point. The theoretical results in Fig. 3 have been calculated by using the Schrödinger equation in tight binding limit Eq. (4). In order to clarify this point an additional explanation has been added to the caption.

Comment:

(7) *I was expecting to see a more drastic difference between the intensity plots in Fig. 4c and Fig. 4d, right panels, since a phase transition occurs between them. Can the authors clarify why these intensity plots look so similar? Also, these panels need a colorbar.*

Response:

The system undergoes two transitions, one related to the topological states (edge property), and one associated with the eigenvalues of the broken/unbroken PT symmetry (bulk property). The first one can be probed by looking at the existence of edge states, while the other one can be probed by observing the standard deviation in the bulk. This is the reason why the topological transition is visible in Fig. 4 but not Fig. 3, while it is the other way around for the broken/unbroken PT transition. In order to clarify this point, we have outlined the different phases in Fig. 3. A colorbar has been added as well.

Comment:

(8) A typo: "the full potential of PT-symmetric in higher dimensions" -> "PT symmetry" or "PT-symmetric systems"

Response:

We thank the referee for spotting this mistake, which has been corrected in the manuscript.

Reviewer #3 (Remarks to the Author):**Comment:**

In this manuscript entitled "Demonstration of a two-dimensional PT-symmetric crystal: Bulk dynamics, topology, and edge states", the authors reported their experimental demonstration of PT-symmetry phase transition and topological phase transition in a photonic waveguide lattice. The waveguide lattice is fabricated in fused silica glass using laser direct writing technique, where parameters of the system, e.g. additional loss, coupling strength, strain strength, lattice structure, can be adjusted. By judiciously choosing parameters, a two-dimensional PT-symmetric waveguides lattice with honeycomb structure is achieved. This PT-symmetric system not only has broken/unbroken PT-symmetry phase transition, but also has topological phase transition due to its special band structure. The merit of this work lies in the 2D lattices which could demonstrate some features beyond what's observed in 1D PT-symmetric structures. Overall the manuscript is well organized.

Response:

We thank the reviewer for his time and effort to read and evaluate the manuscript. Furthermore, we are grateful for the positive reception of our work.

Comment:

1. In Eq. (2), it seems that the unit of the term on the left hand side does not match with that of the right hand side.

Response:

We agree and changed this in the manuscript.

Comment:

2. Eq. (1) and Eq. (2) set up the framework to study *PT* symmetry in two-dimensional photonic crystals, based on which the constraints on the complex refractive index can be derived. Yet in the following discussion, the condition of *PT* symmetry is given with respect to a one-dimensional optical potential, as shown in Eq. (3). It will be helpful if the authors could provide more discussion on the condition of *PT* symmetry for a two-dimensional system.

Response:

We thank the reviewer for pointing out this issue. The extension to a 2D structure is straightforward and consist of a flip in the x and y direction. This is now explicitly stated in the manuscript.

Comment:

3. There is a lack of argument or demonstration explaining why the refractive index potential in this two-dimensional photonic graphene lattice satisfies the condition of *PT* symmetry.

Response:

We agree that this point was not clear in the original version of the manuscript. The explanation given in response to the previous comment, together with the illustration in Fig. 1a, should remedy this flaw.

Comment:

4. In Fig. 3, the number of data points provided is not enough to show a complete trend in which the standard deviation of the output intensity changes. In particular, more data points are needed to show the eventually vanishing behaviour when the system is in the unbroken *PT* regime

Response:

We completely overhauled the figure and added a theoretical graph and its error range resulting from a slight mismatch of injection efficiencies of the two sublattices. Furthermore, the phase transitions are shown in order to allow for a better comparison to Fig. 4. Experimental constraints such as the aforementioned differences of the injection efficiency or slight fluctuations in the measured intensities prevent an observation of vanishing standard deviation in the experiment.

Nevertheless, the standard deviation clearly bottoms out and substantially falls below the zero-strain case. In order to confirm that the standard deviation remains at such low level, we performed an additional experiment with the same loss but significantly higher strain. This additional data point is shown in the figure accompanying this response, and clearly supports our claims. Note that we chose not to include this data point in the figure of the actual manuscript, since the interesting physics is

better captured with the strain ranging between 1 and 3.5. With all those arguments combined we hope to convince the reviewer that our experiments are indeed showing the claimed behaviour.

Comment:

5. In Fig. 3, the experimental data seem to have a slight deviation from the theory. In theory, the standard deviation does not exhibit a sharp decrease to zero after the phase transition point ($\tau=2.316$ as derived from the parameters provided). But in the experiment, the change appears to be abrupt around the phase transition point. The reason for this difference should be provided.

Response:

As pointed out in the previous response, there are several possible sources of error, which have influence on this measurement, such as the injection efficiency mismatch, and fabrication inaccuracies due to laser fluctuations. Especially with the extra figure we hope to convince the reviewer that there is definitely no upward trend but merely fluctuations within the band of measurement uncertainty.

Comment:

6. In this experiment, the PT-symmetric optical potential is realized with an alternating loss, and thus the system exhibits an overall lossy feature. How is the topological phase transition point, as mentioned on page 6, influenced by the global loss of the system?

Response:

Since the loss is global, it can be interpreted as a mere prefactor of the resulting intensity pattern (see Ref [24]). As such, it does not influence the transition into the topological regime.

Comment:

7. On page 7, the authors state that “Moreover, we highlighted the close connection of a PT-symmetry phase transition to a topological phase transition in our graphene lattice.” Except for the conventional argument that “the topological mid-gap state spontaneously breaks PT-symmetry” (page 6), it is not apparent how these two kinds of phase transitions are related. The link between them should be clarified.

Response:

As the reviewer points out, one of the key arguments is that the topological mid-gap state inherently breaks PT symmetry. This is actually also the undisputed main argument for the link between these two phenomena. However, there is indeed also another argument for the connection between PT-symmetric phase transitions and topological phase transitions: Only the existence of a band gap allows the proper definition of a topological invariant for the structure. However, the opening of a band gap in a PT-symmetric structure is associated with the transition between the broken and unbroken PT regime. The way in which the two transitions depend on each other is shown in Fig.4a.

Comment:

8. It will be helpful if the authors could provide more information on the advantageous features and

applications of the two-dimensional PT-symmetric crystal compared to the previous one-dimensional platforms with PT symmetry.

Response:

Since a similar comment was made by reviewer 1, we will give a brief summary, and would like to refer to the more detailed reply starting in the last paragraph of the first page of this response letter: With this work, we show that it is possible to study bulk, as well as edge properties of PT symmetric systems (cf. Fig 3 & 4), hence paving the road for numerous future experiments in this field. Topics of interest are e.g. the interplay of Anderson localisation with PT symmetry, the study of nonlinear effects, like solitons, which behave fundamentally different in 2D, and probably the most important topic, PT-symmetric topological insulators.

Comment:

9 When this system undergoes a PT-symmetry phase transition, it is theoretically predicted that a topological phase transition will be accompanied. More interesting, when the loss difference $\gamma > 0$, these two transitions can be observed at different values of strain τ , which is shown in Fig. 4. But the observed results in the right panel of Fig. 4(b)-(d) are not very clear. The paper says in Fig. 4(c) & 4(d) the light injected is spreading into the bulk due to the absence of edge states, and in Fig. 4(b) the edge states exist thus light stays close to the edge. However, the lattice structure presented in this manuscript only has three sublattices, thus the definitions of “edge part” and “bulk region” of the system is not very clear. In Fig. 4(b), it does not seem like the light is just “remaining at the edge”, one can also see some light penetrates into the bulk. In order to convincingly state that the topological edge states are present, is there a better way to visualize and quantify this effect? It is important to clarify this issue in Fig. 4, because “the demonstration of a non-Hermitian two-dimensional topological phase transition” is a key result of this work, as said in the abstract. It seems a larger system with more sublattices will make it clear that the light in the waveguides is really confined to the edge.

Response:

The lattice consists of 42 waveguides altogether, with about 9 waveguides constituting the “bulk” and remaining 33 representing the “edge”. Even if the edge modes of this array are extending into the bulk, they will still predominantly reside at the edge. Furthermore, one can achieve a large overlap with a possible localised mode, if a single waveguide is excited directly at the edge, as done in the experiment. Since the emergence and disappearance of edge states is a key result of the paper, we decided to quantify the localisation in a more conclusive way. To this end, we calculated the intensity of light within the marked (excited) waveguide, in proportion to the intensity of the entire lattice. This analysis clearly indicates that the localisation in Fig. 4b is, with about 43.5 %, much stronger compared to Fig. 4c and Fig. 4d, which yield values of about 6.3 % and 5.9 %, respectively. These results support the claim that edge modes are present for the parameters Fig. 4b but not for the parameters in Fig. 4c and Fig. 4d. The revised version of Fig. 4. now displays these values above the pictures of the output intensity distributions.

Comment:

10 There are some typos in the manuscript. For example, on the left hand side of Eq. (1), $\partial/\partial z$ should

be corrected as $\partial/\partial t$, i.e., the partial z should be partial t. In Figure caption 4, "(b)" is repeated twice where "(c)" and "(d)" should be used.

Response:

We thank the referee for pointing out these mistakes, which have been corrected in the revised version.

Reviewers' Comments:

Reviewer #1:

Remarks to the Author:

The authors have revised the manuscript to my satisfaction and now recommend it for publication.

Reviewer #2:

Remarks to the Author:

I believe the authors have done a very good job in addressing the issues raised in my previous review, as well as the comments of the other reviewers. The revised manuscript is definitely improved from the original version, and I'm glad to recommend it for publication in Nature Communications.

Reviewer #3:

Remarks to the Author:

The authors have carefully addressed the issues in my previous report. Now I think the manuscript is ready for publication in Nature Communications.

Reviewers' comments:

Reviewer #1 (Remarks to the Author):

Comment:

The authors have revised the manuscript to my satisfaction and now recommend it for publication.

Response:

We thank the reviewer for this positive feedback and the time and effort to evaluate the manuscript.

Reviewer #2 (Remarks to the Author):

Comment:

I believe the authors have done a very good job in addressing the issues raised in my previous review, as well as the comments of the other reviewers. The revised manuscript is definitely improved from the original version, and I'm glad to recommend it for publication in Nature Communications.

Response:

We thank the reviewer for this positive feedback and the time and effort to evaluate the manuscript.

Reviewer #3 (Remarks to the Author):

Comment:

The authors have carefully addressed the issues in my previous report. Now I think the manuscript is ready for publication in Nature Communications.

Response:

We thank the reviewer for this positive feedback and the time and effort to evaluate the manuscript.